# The Olive Grove Landscape as a Tourist Resource in Andalucía: Oleotourism

**Mª Genoveva Dancausa Millán** [1], **Javier Sanchez-Rivas García** [2] **and Mª Genoveva Millán Vázquez de la Torre** [3,*]

1   Department of Statistics, University of Cordoba, Puerta Nueva s/n, 14071 Córdoba, Spain; z62damim@uco.es
2   Department of Economic Analysis and Political Economy, University Seville, Ramon y Cajal 1, 41018 Seville, Spain; sanchezrivas@us.es
3   Department of Quantitative Methods, Universidad Loyola Andalucía, 14004 Córdoba, Spain
*   Correspondence: gmillan@uloyola.es

**Abstract:** Andalucía, located in the southern region of Spain, is the world's largest producer of olive oil. It is home to over 70 million olive trees, which shape the distinctive landscape of the olive groves typical of the Mediterranean Basin. This research focuses on analyzing the olive grove landscape and the rich culture associated with the olive tree as a tourist activity known as oleotourism. This form of tourism would attract an increased number of tourists and generate funds for the preservation of the olive tree heritage if UNESCO declared it a World Heritage Site, similar to other cultural landscapes. Furthermore, it would help diversify the region's tourism industry, which has traditionally focused on sun and beach tourism. This study is a continuation of previous research initiated by the authors in 2017. It is observed that the profile of oleotourists before and after the COVID-19 pandemic has not varied significantly, although there has been an increase in the number of people engaging in this type of tourism, as evidenced by the results obtained with the SARIMA model. The SARIMA model is used for forecasting and analyzing time series data. These findings indicate that investing in this tourism activity would be profitable for local residents, making it a sustainable form of tourism that is compatible with the environment and the local community.

**Keywords:** landscape; olive grove; heritage; Andalucía; oleotourism; SARIMA





## 1. Introduction

The landscape of a region is the visual and aesthetic result of the interactive impact on the territory of climatic, relief, water, soil, natural flora and fauna factors, and human actions. The outcome of this interaction is ultimately a specific spatial arrangement of agroecosystems, which is a characteristic of each territory, constituting its most perceptible dimension [1]. There exists a subject observer and an object observed, with the latter primarily highlighting its visual and spatial qualities. Therefore, the landscape is a "fundamental component of the natural and cultural heritage of many countries, regions, and districts, a key element of individual and social well-being, and a favorable resource for economic activity capable of generating employment, income, and thus, essential for population retention in rural areas and promoting the development of those areas.

Rural landscapes are the result of a close relationship between agriculture and landscape; however, the traditional links between the two have been renewed and reoriented in recent decades due, among other reasons, to the academic and normative acceptance of the approaches of the European Landscape Convention [2]. The European Landscape Convention (ELC), which has been ratified by most of the member countries of the Council of Europe, aims to establish a framework for the protection, management, and planning of European landscapes. Its ultimate objective is to preserve and enhance their quality. The strategies it proposes encourage the involvement of the public, institutions, authorities, and local, regional, national, and international actors in public decision-making processes. The Convention recognizes all forms of European landscapes, natural, rural,

urban, and peri-urban, both emblematic and ordinary, and those that have been degraded, as a non-renewable resource.

Thus, the importance of the landscape dimension of the territory is beginning to be recognized, as reflected in certain international declarations that emphasize the need to preserve landscapes as an essential element of world culture (UNESCO World Heritage Convention, 1972, as amended in 1992; Mediterranean Landscape Charter, 1992; European Landscape Convention, 2000; European Union Territorial Agenda, 2020, 2011). The territory itself constitutes a cultural and economic resource of the highest order, which leads to a new concept, that of territorial heritage [3], referring to both its physical–natural disposition and the heritage resources available in each area, including intangible ones that reinforce identity and confer greater competitiveness on territories. The territorial dimension possesses specific resources that mark differences and enhance competitiveness in the current socio-economic context. As indicated by Millan et al. [4], any element of a territory can serve as a potential factor for development: local products, traditional ways of obtaining them, landscapes, accommodations, architecture, significant historical events, flora, fauna, rivers and aquatic spaces, topography, gastronomic traditions, music, festivals, rituals, images, idioms, knowledge, agricultural activities, etc.

Increasingly, certain intangible elements and activities related to territorial identity are being valued more, revealing themselves as fundamental in presenting a new conception of development in rural areas. Their value corresponds to their connection to local culture and heritage. This is particularly evident in everything surrounding the world of olive trees, olive oil, and the landscapes of the olive groves in Andalucía, Spain.

The term "Cultural Landscape", for UNESCO, encompasses a diversity of manifestations of interaction between humans and their natural environment. Some cultural landscapes reflect specific techniques for sustainable land use and take into consideration the characteristics and limits of the natural environment in which they are established, as well as a specific spiritual relationship with nature. Thus, the protection of cultural landscapes contributes to modern techniques for sustainable land use and the preservation or enhancement of the natural values of the landscape. However, there are cultural landscapes not linked to agricultural activity, such as those created by mining activity, where slopes are devoid of vegetation. In Spain, there are 27 mining landscapes [5,6], or landscapes generated naturally, forming ecosystems like the Amazonia [7].

For some time now, the concept of cultural landscapes, linked to human intervention, has been receiving increasing attention from researchers and planners [8–11]. The connection between the concepts of heritage, cultural landscape, and the historical shaping of the territory is particularly relevant, with agricultural landscapes related to the rural world standing out.

As indicated by Aladro-Prieto et al. [12], the preservation of agricultural heritage systems of great aesthetic and cultural significance, as well as the associated traditional agricultural practices, is necessary. These practices allow for the production of high-quality food appreciated by consumers who are willing to pay a higher price for products that maintain a close connection with the territory of origin, such as olive oil and olive landscapes. These landscapes hold value due to their aesthetic beauty, knowledge of production methods, history, culture, and traditions, which cannot be replicated by competitors. Therefore, focusing on these elements is crucial to maintaining traditional, productive, vital, and profitable landscapes [13,14]. Failure to do so would result in the disappearance of these landscapes, as has occurred for many cultural landscapes throughout Europe when agricultural production was no longer economically viable. Hence, the importance of recognition in order to preserve them.

This paper analyzes oleotourism (olive oil tourism) in Andalucía, which can contribute to the conservation of olive landscapes in the region and increase the income of residents in rural areas engaged in olive oil production. In essence, this tourism activity is considered another aspect of the multifunctional nature of these landscapes, which can contribute to their preservation while diversifying the utilization of resources related to olive culture

and generating income from their exploitation. All of this will contribute to enhancing the region's economic and social development. Through these tourism initiatives, economic activities in rural Andalucía can be diversified, linking the valorization of territorial heritage (olive landscapes) with the promotion of rural and regional development.

## 2. The Olive Grove Landscape in Andalucía and Tourist Activity

Some agricultural landscapes receive special attention from a heritage perspective, such as traditional irrigation systems, dehesa (a type of agroforestry system), cereal plains, vineyard landscapes, and olive grove landscapes. According to Delgado et al. [15], the olive grove is a constant feature of the diverse Mediterranean landscapes, with its dominant perennial, evergreen tree species, relatively open canopy, and varied herbaceous and floral understory creating a vibrant image with seasonally changing hues.

The olive grove landscape is not a recent phenomenon; the first archaeological references to the product obtained from the olive tree, olive oil, can be found in Ancient Egypt. The Egyptians attached great importance to oil due to its scarcity, as the olive variety cultivated there had low yields. It was primarily used in pharmaceutical or cosmetic products rather than in cooking. Trade between the Egyptians and the Greeks spread olive oil throughout Ancient Greece. In the early Olympic Games, champions were rewarded with an olive branch as a symbol of their victory, highlighting the significance of the olive grove and its fruit during that period. Writers like Homer mentioned olive oil as "liquid gold" in their works, such as *The Iliad*. During the time of the Roman Empire, its inhabitants adopted the customs, traditions, and techniques of the Ancient Greeks and refined them. They utilized Greek knowledge of olive cultivation to optimize olive oil production and learned the secrets of olive oil processing, notably using the "molea olearia," an animal-driven oil mill that increased oil production [16]. The expansion of the Roman Empire and its arrival on the Iberian Peninsula (Spain–Portugal) led to the continuation of olive cultivation. Even today, ancient olive trees from the Greek and Roman eras can still be visited in Andalucía. Olive oil, along with wine, have remained fundamental elements of the agrarian economy, persisting from ancient times to the present day.

The consideration of agricultural systems in rural or regional development processes, beyond productivity and mere financial profitability, has been the subject of various studies due to the increasing societal sensitivity toward the consideration and valorization of certain externalities, particularly those of an environmental nature. In general, the new functions attributed to these agricultural systems are linked, according to Laurens [17], to the emergence of new products, new rights, and new public goods in which agricultural systems are directly or indirectly involved. Characteristics such as the sustainability [18–20], productive quality [21,22], balance, efficiency, total economic value [23,24], environmental interaction level [25], and socio-cultural aspects [26] of these systems should be taken into account when assessing their current functioning and potential transformations, thereby determining their suitability within frameworks of rural and/or regional development processes.

Furthermore, some studies analyze not only the aesthetic aspect of agricultural systems but also the tertiary economic activities they generate or participate in, such as rural tourism [27–32], or specifically, olive tourism [33–37], as activities that complement those considered essential for the local economy. Ultimately, everything points toward the concept of the multifunctionality of agricultural systems, which allows the olive grove landscape and olive culture to be transformed into tourist attractions through activities such as olive tourism, generating additional income alongside agricultural activities.

The tourism sector has historically been one of the priority sectors for economic activation and growth worldwide. International tourism represents 7% of total global exports and 30% of service exports. In Spain, until the year 2019, tourism accounted for 12.4% of the GDP, making Spain a global tourism leader and one of the preferred destinations for travelers from around the world. However, due to the COVID-19 pandemic and mobility restrictions, this contribution significantly declined in 2020, making it the worst year in history for the tourism sector. In 2021, this fall continued to reflect data

significantly lower than the values recorded for 2019, but in 2022, a recovery of the sector was observed, with figures very similar to those of 2019.

During this period (2020–2022), domestic tourism has helped sustain inland tourism businesses, as Spanish tourists have sought destinations within the country. Inland areas have gained great importance as tourists have sought safe and healthy destinations. Rural inland areas have become recognized as safe places, and tourists have appreciated the new tourism offerings available, including olive tourism. Olive tourism encompasses activities such as visiting olive grove landscapes to appreciate ancient olive trees, visiting olive oil mills to learn about the oil production process, and participating in olive oil tastings to experience different oil varieties, among other activities.

In the international arena, this tourism sector is subject to continuous changes due to the increasing number of factors that influence its formation as an independent economic engine. The traditional tourism model (mass seasonal tourism) is undergoing transformation, giving rise to multiple tourism realities characterized by the diversity of segments and products. In this complex and ever-changing scenario, the Andalucían region is also immersed, a destination that has great attractiveness. The Autonomous Community of Andalucía, located in the south of Spain, is characterized by being a tourist area of great singularity, where different degrees of development and tourism exploitation models coexist, with the strong presence of the sun and beach proposal, as it has 910 km of coastline and 373 beaches. The region is strongly linked to this type of tourism, mainly due to its privileged geographical position and the favorable climate it enjoys for much of the year [38]. It offers 113,700 hotel rooms, with an average occupancy rate of 53.89%, although private tourist accommodations are gaining more strength. As for the personnel working in hotels, there are 34,976 individuals.

In the year 2022, the Andalucían region received 30.7 million tourists, with Malaga being the province that received the most tourists, 8.3 million (27.1%), followed by Cadiz with 5.4 million tourists (17.6%), both being coastal provinces. The origin of the tourists visiting Andalucía was as follows: 36.9% were from the same region, 31.8% from the rest of Spain, 24.3% from the European Union, and 7% from the rest of the world. The average stay was 6.4 days, 8.4 days for European Union tourists, and 11.6 days for the rest of the world. Tourists from non-European Union countries spent the most during their stay, with an average of EUR 98.1 per day, followed by European Union tourists with EUR 82.3, while Andalucíans spent EUR 62, and the tourism was concentrated mainly from April to October [39]. To address the seasonality of this type of tourism, the Andalucían government has designed the General Plan for Sustainable Tourism in Andalucía META 2027, which aims to improve the management of the socio-economic activity of tourism by its main stakeholders within a framework of social, economic, and environmental sustainable development, promoting a competitive and entrepreneurial model of quality, intelligence, equality, and inclusiveness, based on human resources and the identity value of the Andalucía destination. This plan is based on 11 objectives:

1.  Consolidate the role of tourism as a vehicle for sustainable development and the creation of stable, qualified, and high-quality employment for the Andalucían economy.
2.  Advance toward a new tourism management model whose fundamental pillars are environmental, economic, and social sustainability, particularly benefiting the inland towns where the average income per capita is very low. This includes creating new tourism offerings such as olive tourism that allow for additional income through this tourism activity and prevent the migration of young people from rural areas to cities.
3.  Guarantee a model of tourism development based on integration and excellence, as well as inclusive, accessible, and multigenerational tourism.
4.  Ensure greater coordination of tourism planning with similar tools of the Andalucían Regional Government and other national and international organizations.
5.  Optimize the profitability and competitiveness of the Andalucían tourism sector through excellent services and tourist destinations.

6. Consolidate the competitive transformation of the Andalucían tourism industry, where innovation, continuous digital adaptation, and an emphasis on tourism intelligence constitute factors of competitiveness for our tourism sector.
7. Promote the development of new strategies for academic and professional training and support for tourism entrepreneurs.
8. Enhance strategies aimed at reducing the seasonality of tourism activity by creating products and developing segments that can be implemented throughout the year, contributing to territorial cohesion. Olive tourism with visits to the olive grove landscape would be included within this objective.
9. Develop a marketing policy that promotes better commercialization of products and destinations, responds to the motivations of an increasingly diverse range of visitors, and highlights the unique characteristics of each territory under the umbrella of the Andalucía brand.
10. Consolidate the regeneration of the Andalucían tourism sector, focusing on a safe and healthy destination.
11. Ensure legal security in the practice of tourism activities.

Therefore, the META plan aims to accomplish the following:

- Increase the contribution of tourism to the Andalucían economy compared to pre-health crisis levels, improving tourist income by an average cumulative annual growth of 3%.
- Generate quality, stable, and equal employment in the Andalucían tourism sector, with a cumulative annual average growth of 2.5% in the number of employees. This seeks to achieve the Sustainable Development Goals 5 (gender equality) and 8 (decent work and economic growth).
- Optimize the process of continuous technological adaptation and transformation in the Andalucían tourism sector, especially in rural and inland areas.
- Contribute to a more homogeneous territorial and temporal distribution of tourist flows, reducing seasonality by at least 2% in the cumulative annual average rate.
- Promote and improve the comprehensive sustainability management of Andalucían tourist destinations. This includes achieving the goals of Affordable and Clean Energy (Goal 7), Clean Water and Sanitation (Goal 6), Responsible Production and Consumption (Goal 12), Climate Action (Goal 13), Life Below Water (Goal 14), and Life on Land (Goal 15).

Andalucía has not remained unaffected by the evolution of the tourism sector, and although its tourism offerings are still associated with the classic segment of sun and beach, in recent years alternative proposals have emerged. These include sports tourism, adventure tourism, cultural tourism, urban tourism, rural tourism, cruise tourism, business tourism, gastronomic tourism, industrial tourism, language tourism, health and beauty tourism, and nature tourism, among others.

The variety of landscapes in the rural areas of Andalucía, such as the dehesa and olive groves, allows these elements to be utilized not only from an agricultural perspective but also from a tourism perspective (rural and nature tourism). In Spain, agriculture has been decreasing its contribution to the national GDP in recent decades. However, the olive oil sector still represents an important source of wealth, not only economically but also culturally and in terms of identity when it comes to external perceptions. The need for the Spanish countryside, especially in Andalucía, to find new channels for economic diversification has raised the possibility, considering recent tourism trends, of tapping into a relatively unexplored segment related to olive oil production and consumption activities (known as olive tourism, which can be categorized as rural tourism, industrial tourism, or gastronomic tourism). It is worth noting the unequal impact of the crisis on each sector of the productive fabric, with behaviors reversing in recent years. When disaggregating the economy into the four main sectors of activity, it can be observed that one sector, the services sector (1.55%), experienced strong growth in 2021, while the agriculture sector declined by 4.36% (Figure 1).

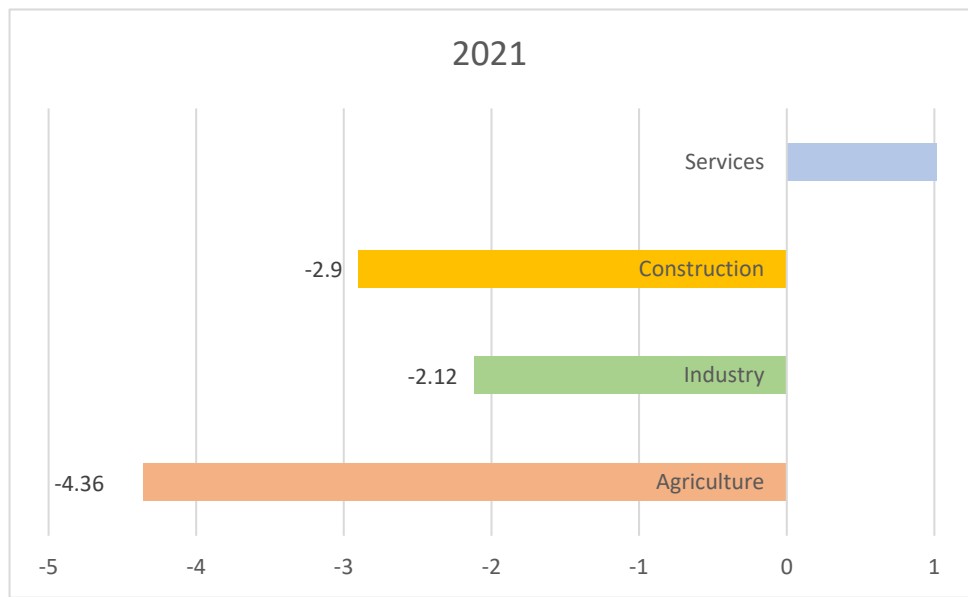

**Figure 1.** Contribution of each economic sector to Spain's GDP in 2021 (in %). Source: Own elaboration from [40].

Therefore, although agricultural production continues to be the main economic activity in rural Andalucía, it cannot be considered the sole pillar on which its development is based. In the current context, the role of heritage resources linked to a unique and irreplaceable territory is strengthened. This is the case with the landscapes of the olive groves in Andalucía, where the territory goes from being considered a passive agent or mere physical support for development processes to becoming an active agent in itself. The possibility of transforming certain territories into tourist settings that facilitate development will lead many rural regions of Andalucía to see tourism as almost the only alternative to overcome economic stagnation and demographic collapse, especially in mountainous areas with difficult access or problems in generating development processes.

For many areas of the Andalucían community, it is crucial to harness those territorial resources, especially those related to tourism, that can contribute to their development. Visiting the olive grove landscape as an activity within olive tourism is a resource to be utilized, given the thousands of hectares of olive groves in the Andalucían region.

In Table 1, we can observe the cultivation area of olive groves in Spain, which is 2.77 million hectares, of which 1.67 million hectares are located in the Andalucían region (Figure 2). This represents over 60.3% of the total olive grove area in Spain, which can be utilized not only as an agricultural resource but also as a tourist resource.

**Table 1.** Olive grove area in hectares and % of the total.

| Region | Surface Area of Olive Grove (Hectares) | % |
|---|---|---|
| Andalucía | 1,673,071 | 60.38 |
| Aragón | 60,332 | 2.18 |
| Balearic Islands | 9126 | 0.33 |
| Canary Islands | 435 | 0.016 |
| Castilla León | 6986 | 0.252 |
| Castilla la Mancha | 449,387 | 16.22 |
| Catalonia | 114,350 | 4.127 |
| Valencian Community | 95,695 | 3.454 |
| Extremadura | 288,692 | 10.426 |

**Table 1.** *Cont.*

| Region | Surface Area of Olive Grove (Hectares) | % |
|---|---|---|
| Galicia | 52 | 0.002 |
| Madrid Community | 29,621 | 1.069 |
| Region of Murcia | 29,032 | 1.048 |
| Navarre Community | 9922 | 0.358 |
| Basque Country | 322 | 0.013 |
| La Rioja | 3475 | 0.125 |
| **Total** | **2,770,498** | **100** |

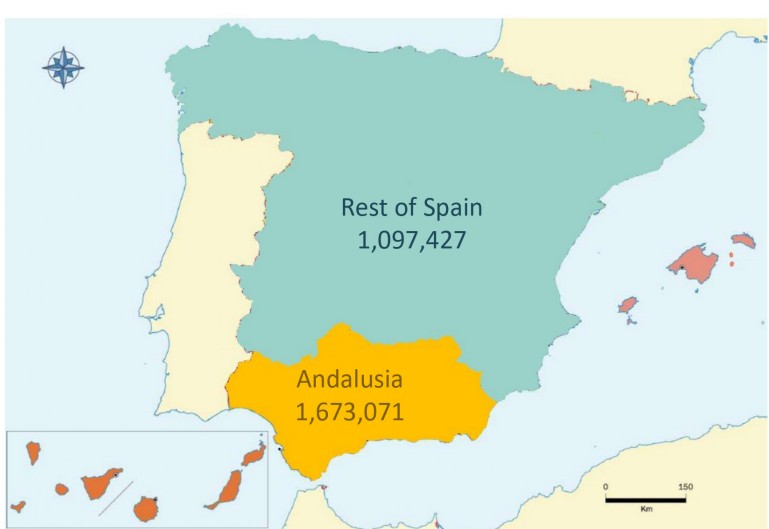

**Figure 2.** Surface area of olive groves in Andalucía–rest of Spain. Source: Own elaboration from [41].

In recent decades, market demands have led to the intensification of olive cultivation through the industrialization of its management practices. Mechanization and the addition of not only water but also other inorganic external inputs have succeeded in breaking the traditional biennial bearing (alternation of good and bad harvest years) of the olive tree, which is problematic due to its potassium metabolism. This response to the needs of an expanding market with inflexible periodicity has resulted in questionable management and practices in those beautiful and identity-rich landscapes, leading to two main negative effects:

- Increase in erosive processes in the Mediterranean context—already notoriously affected by its geomorphology and irregular climatology—where both the wild olive groves (acebuchal) and cultivated olive groves play and should continue to play a significant role in supporting slopes and hillsides (Figure 3), instead of becoming promoters of gullies, loss of fertile soil, and desertification. Losses of nearly 90 tons per hectare per year have been documented, far exceeding any sustainable level of soil loss [42].
- Noticeable loss of biodiversity in olive groves that, due to the addition of chemical products, become devoid of their colorful vegetation cover and cease to be substantial habitats for diverse wild flora and fauna [43–45].

These two negative effects occur as a result of attempts to increase productivity. In 1890, the average olive production per hectare was 1000 kg of olives, whereas in the 21st century, it approaches an average of 4000 kg of olives per hectare, and in some areas like the province of Málaga, it reaches 6000 kg per hectare. This agricultural intensification can impact the traditional olive grove landscape, making it necessary to protect it.

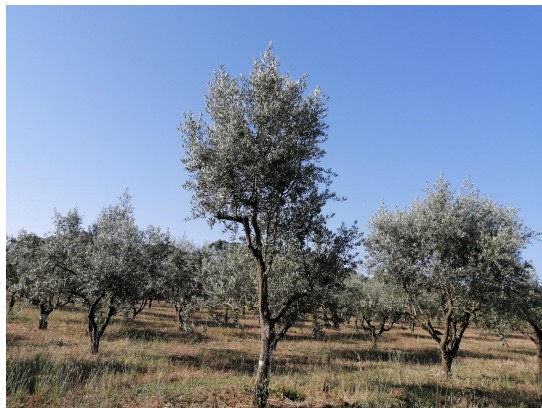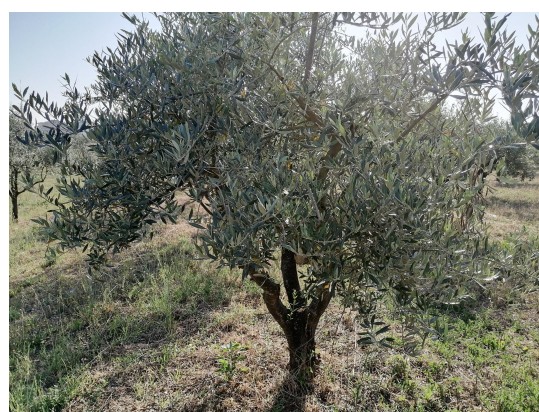

**Figure 3.** Abandoned olive groves (acebuchal). Source: Authors.

### 3. The Olive Grove Landscape as a World Heritage Site Declaration

For several years, both entrepreneurs and public entities have expressed the exceptional universal value of "one of the great crops that are still awaiting inscription on the World Heritage List". Despite the importance of this crop and its immense cultural significance as a symbol of the Mediterranean region and the revered status of the olive tree, its fruit, and oil in ancient times, including in the three monotheistic religions of the Mediterranean, efforts have been made to promote this landscape internationally as the "sea of olive trees".

To achieve this, a proposal has been made for "The Olive Grove Landscape in Andalucía" to be the Spanish candidate for World Heritage status in 2023. In the candidate dossier, several significant cultural landscape areas are identified:

■ Four areas linked to the olive specialization of the 19th century: Campiñas de Jaén, Subbética Cordobesa, Sierra Mágina, and Hacienda de La Laguna—Alto Guadalquivir.

■ Three areas referring to the Enlightenment-era olive groves, represented by Montoro and its surroundings, and the Haciendas of Seville and Cadiz, associated with the 16th to 18th centuries.

■ An area related to the olive groves of the medieval-Islamic period in the Lecrín Valley (Granada).

■ The area of Valle del Segura linked to the olive groves of the 13th to 15th centuries.

■ The Astigi-Bajo Genil area (Écija) connected to the olive groves of the Roman period, from the 1st to the 3rd centuries.

■ The area of Periana and Alora in Malaga, representing the early cultivation practices of the olive tree.

The olive grove landscape is a paradigmatic cultural landscape that seamlessly integrates tangible and intangible aspects. It has given rise to a prototypical landscape, the sea of olive trees (Figure 4), whose lines extend infinitely and encompass an incredible and impressive heritage of architecture, art, history, ethnography, archaeology, and industry. It is an exceptional witness to a form of exploitation that dates back millennia in the history of humanity and is indissolubly linked to the Greco–Roman culture that emerged around the Mediterranean, perhaps being its greatest hallmark of identity.

If the olive grove landscape is declared a World Heritage Site, this designation will allow for the identification, protection, conservation, and transmission of a unique and extraordinary cultural and natural heritage to future generations. In Andalucía, the history of an unparalleled olive specialization in the world is concretized, and the components of this dossier manifest it, making it tangible in its cultural values, its continuity over time, and its persistence as a productive and living crop that sustains thousands of families in our land, which must be maintained and ensured for the future based on the principles of sustainability.

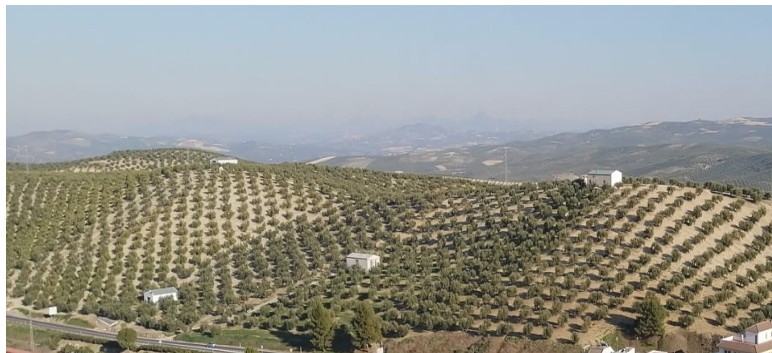

**Figure 4.** The olive grove landscape. Source: Authors.

The olive tree, the olive grove, and its oil are fundamental pillars of Mediterranean culture deeply rooted in Andalucía. The immense olive forest that shapes the Andalucían landscape, particularly in the province of Jaén, is not just an example of beautiful scenery; rather, it is a source of wealth thanks to the oil it produces, along with other derivatives extracted from this plant with almost miraculous properties. Aware of these values, numerous entities of all kinds have initiated a proposal for the Olive Grove Landscapes of Andalucía to be declared World Heritage by UNESCO as an Agricultural Cultural Landscape.

The olive grove in Andalucía has shaped and continues to shape a unique landscape that provides life and culture in the rural territories where it is located. It forms the basis of traditional, social, heritage, economic, gastronomic, touristic, and labor-related aspects, giving rise to a particular way of understanding and experiencing life and attachment to the land since ancient times and throughout the different stages of our evolution. Thus, the history of the olive grove is inseparably linked to the history of Andalucía.

It is a forest composed of more than 70 million trees spread over more than 1.67 million hectares, making Andalucía the largest producer of Extra Virgin Olive Oil worldwide, accounting for 30% of the total production, in addition to 20% of table olives. It constitutes the main economic activity for over 300 Andalucían municipalities, providing more than 22 million workdays per year.

Currently, Spain has various landscapes that have been declared World Heritage Sites, such as the Cultural Landscape of Aranjuez, the Cultural Landscape of the Sierra de Tramontana, Las Médulas in León, Monte Perdido in the Pyrenees, and the Palm Grove of Elche (Figure 5).

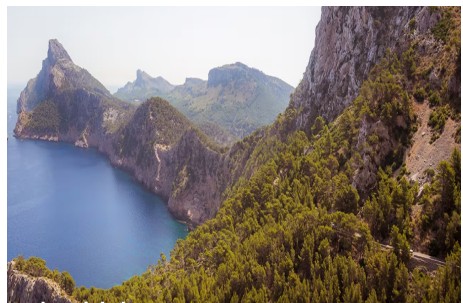 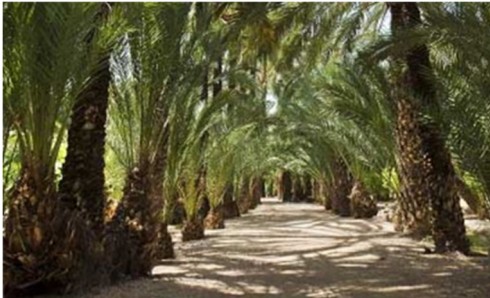

**Figure 5.** Cultural landscape of the Sierra de Tramontana and Palm Grove of Elche. Source: Authors.

The olive grove landscapes in Andalucía, when considered as a whole, exhibit a significant diversity in structure, agronomic practices, and productivity that persists, albeit to varying degrees, across different territorial levels (provincial, regional, municipal). Guzmán [46], using cartographic overlay methods with both analog and digital mapping, has carried out a multivariate contrast that allowed for the territorial classification of olive groves in Andalucía. Consequently, various types of Andalucían olive grove landscapes can be distinguished based on the characteristics of the physical environment: countryside

olive groves, olive groves on sedimentary hills, olive groves on plains and metamorphic slopes, olive groves on hills and plains with alluvial deposits, olive groves on plains with diluvial deposits, olive groves on endogenous plain areas, olive groves on semi-arid interior meadows, olive groves in a mosaic of interior basins, olive groves on plains, hills, and hills with varied lithology, and olive groves in mountainous coastal areas.

Therefore, it is necessary to recognize the nuances of olive cultivation, some of which are quite significant and decisive, often reflecting profound agronomic and territorial differences. Despite this diversity, many of these landscapes are recognized for a set of associated common values (the multifunctional character of Andalucían olive grove landscapes):

➢ Aesthetic values: The capacity of a landscape to evoke a particular sense of beauty. Landscape values form an inseparable set with other cultural elements. In this case, it refers to olive grove areas or environments with outstanding visual beauty.

➢ Ecological values: Factors or elements that determine the quality of the natural environment. An olive grove is not composed solely of olive trees; it also encompasses associated resources such as soil, water, other plants, and the fauna that inhabit it. In this sense, olive cultivation stands out for its adaptation to the Mediterranean climate, its efficient water usage, and its occupation of generally unsuitable land for other crops, often constituting environmentally and aesthetically valuable areas that serve as refuges for unique flora and fauna.

➢ Productive values: The capacity of a landscape to generate economic benefits by transforming its elements into resources.

➢ Historical values: Olive groves preserve a centuries-old cultural legacy that reflects the most relevant imprints left by different civilizations on this landscape throughout history.

➢ Social use values: The utilization of a landscape by individuals or specific communities. Olive groves are agricultural landscapes that, unlike spaces shaped by nature (cliffs, ravines, escarpments) or intentionally designed by humans (parks and gardens), combine physical appearance and functionality in an indissoluble manner.

➢ Symbolic and identity values: The identification of a particular community with a landscape.

All these values are components of Andalucía's authentic territorial capital. Being recognized as a World Heritage Site would be a true catalyst, increasing the number of companies dedicated to olive oil tourism, attracting more visitors, promoting the recognition and appreciation of Extra Virgin Olive Oil in markets, and ultimately, helping the Andalucían olive grove gain universal recognition from a heritage, ethnological, and environmental perspective.

For the olive sector, if UNESCO were to declare the landscape of the Andalucían olive grove a World Heritage Site, the actors involved in the production of Extra Virgin Olive Oil, and other related activities, would benefit economically. The distinction from this organization would lead to the implementation of positive measures, such as an increase in support from the Common Agricultural Policy, a better defense of prices, and the promotion of population retention in the territory. Additionally, it would serve as an excellent international showcase, allowing for better promotion of the product's tradition and history. The close connection of Extra Virgin Olive Oil with the Mediterranean diet would stimulate the modernization of mills and irrigation systems and encourage generational succession, as the olive grove represents a way of life that guarantees the livelihoods of farmers and sustains a socially relevant environment from a quantitative perspective. Moreover, it should not be overlooked that the environmental sustainability aspect implies the preservation of a significant forested area. With increasingly utilized production methods that do not harm the environment, appropriately managed agricultural systems can contribute to rural and regional development in a broad sense.

## 4. Materials and Methods

Currently, Andalucía is considered the largest olive-growing region in the world, with olive oil production representing one of the main economic activities in the region. In 2021,

the Andalucían olive grove covered 1.6 million hectares, with a production of 587,000 tons of olive oil (Table 2). There were 169,459 olive farms and the production value for the 2021–2022 campaign reached EUR 2.240 billion. Additionally, this crop is one of the highest job generators per unit of land, earning it the nickname of a "very social crop." According to the agricultural census, olive cultivation accounts for around 32% of the workforce in the entire agricultural and livestock sector in Andalucía.

**Table 2.** Olive oil production and organic olive oil production (Andalucía).

| Andalucía | Area (Hectares) | Olive Oil Production | Variation from Previous Campaign | Organic Olive Oil Production | Variation from Previous Campaign |
|---|---|---|---|---|---|
| Almería | 22,026 | 10,000 | −25.7 | 300 | −55.5 |
| Cádiz | 30,845 | 9000 | −19.2 | 200 | −34.8 |
| Córdoba | 371,134 | 158,000 | −47.3 | 8000 | −44.8 |
| Granada | 206,694 | 70,000 | −41.4 | 1000 | −45.7 |
| Huelva | 35,749 | 10,000 | −18.5 | 3100 | −18.4 |
| Jaén | 588,252 | 200,000 | −60 | 1200 | −46.8 |
| Málaga | 141,405 | 40,000 | −30.4 | 400 | −31.4 |
| Seville | 242,215 | 90,000 | −35.2 | 2500 | −48.7 |
| Total | 1,638,320 | 587,000 | −49.1 | 16,700 | −42 |

The cultivation of olive trees is, in relative terms, the most economically profitable activity, with better social performance, at least when compared to other alternative uses (agricultural or forestry) of the land. In fact, any circumstance that promotes the abandonment of olive cultivation would have significant negative socio-economic effects: loss of income for farmers, decrease in the wealth of the population as a whole, and an increase in the unemployment rate.

The provinces of Jaén, Córdoba, Sevilla, Granada, and Málaga (Figure 6) make up the so-called "olive axis", with the first two provinces accounting for almost 60% of the Andalucían olive groves.

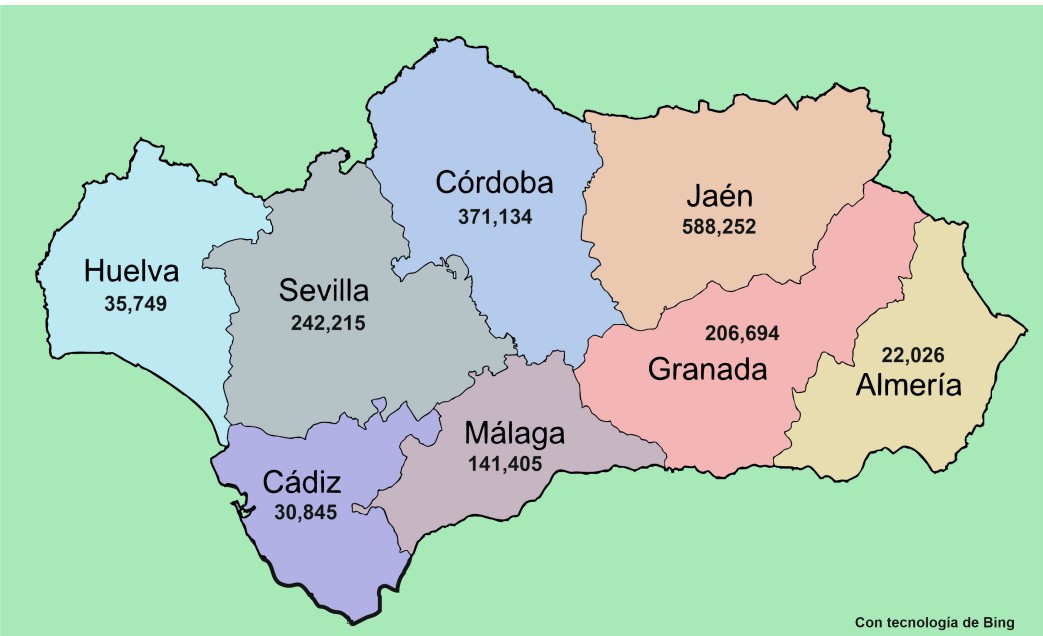

**Figure 6.** Olive grove surface area by province (hectares) in Andalucía in 2022. Source: Own elaboration from [47].

In the province of Jaén, olive groves occupy 46.19% of the agricultural land (Figure 7), with cases in some regions and municipalities where the presence of other crops is minimal or has disappeared completely [48].

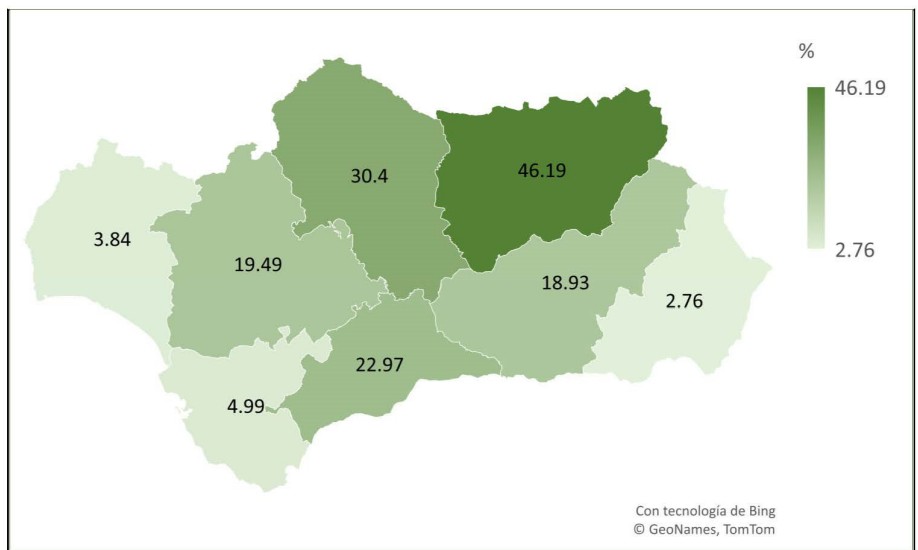

**Figure 7.** Percentage of land dedicated to olive cultivation compared to the total agricultural land by province in Andalucía. Source: Own elaboration from [47].

In recent years, and especially after the COVID-19 pandemic, where there have been changes in consumer tourism behavior, a tourism activity related to the world of olive groves, gastronomy, and rural areas called "oleotourism" has been developing.

This research is a continuation of a study carried out in 2017 on the demand for oleotourism in the Andalucían region [49]. In this second phase, the impact of COVID-19 on the demand for oleotourism is analyzed.

The scope of the study is oleotourism in Andalucía. A survey was conducted among a population composed of tourists who visited any of the olive oil denominations of origin that make up the olive grove axis in 2022. The objective was to understand the profile of the oleotourist. For this purpose, a questionnaire consisting of 30 questions divided into 5 sections was administered (Table 3). The first section collects personal information about the tourists (age, gender, education level, marital status, etc.). The second section contains information about the route taken (how they learned about the olive oil route, whether the route met their expectations, what improvements they would suggest, if they specifically came for the gastronomic route, etc.). A third section focuses on the motivation for engaging in oleotourism (what motivated them to take the gastronomic route and their use and consumption of olive oil at home). A fourth section deals with evaluation (their assessment of the services received during the route, the price of the trip, hospitality and treatment received, etc.). The fifth section is related to olive oil (their knowledge about olive oil, its use, personal preferences regarding olive oil, etc.). The access by the surveyors to the olive oil route/PDO/PGI and the conducting of interviews with tourists were authorized by the managing body and owner of the PDOs/PGIs. Prior to completing the questionnaire, tourists were informed of the academic purposes and the anonymity of their answers. Verbal consent was obtained to participate in the questionnaire. The anonymity of the visitors to the olive oil route/PDO/PGI was guaranteed at all times.

**Table 3.** Technical aspects of the survey.

| | **Demand Survey** |
|---|---|
| Population | Tourists over 18 years of age who visited an olive oil route/PDO route in Andalucía |
| Sample size | 416 |
| Sampling error | ±4.2% |
| Confidence level | 95%; p = q = 0.5 |
| Date of fieldwork | March 2022 to October 2022 |

With the information obtained in the survey, the following were carried out:

1.  A univariate descriptive analysis was conducted to determine the profile of oleotourists, with the objective of identifying if there are significant differences between tourists.
2.  A bivariate analysis was conducted using contingency tables to identify whether there is an association or independence between two variables, using the $\chi 2$ statistic (where $H_0$ is that the analyzed variables are independent and $H_1$ is that the analyzed variables are related). The aim of said analysis was to determine the associations between variables, thus allowing for the identification of the profiles of gastronomic tourists.
3.  A bivariate analysis was conducted using a test to compare means regarding some evaluations obtained before and after the COVID-19 pandemic.
4.  A SARIMA (Seasonal Autoregressive Integrated Moving Average) model was applied to forecast the demand for oleotourism in the year 2023, aiming to understand the evolution of this tourism segment and identify its challenges and opportunities. The SARIMA model (used in previous studies of tourists, such as those by Lim [50] to predict tourist demand in Macao after the COVID-19 pandemic; Yang and Zhang [51] to predict the tourist demand in 29 Chinese regions; Petrevska [52] to predict the tourist demand in Macedonia; and Zhang [53] to predict tourist occupancy in a hotel) was used to predict the potential demand for oleotourism in Andalucía based on a sample (170 observations) collected from January 2009 to February 2023. SARIMA models, popularly known as the Box–Jenkins (BJ) methodology, analyze the probabilistic, or stochastic, properties of economic time series themselves [54]. In this case, this was the number of oleotourists in Andalucía.

## 5. Results

### 5.1. Univariate Descriptive Analysis

Table 4 shows the main variables expressed in percentages and their comparison across the three analyzed periods. An increase in the age of visitors can be observed. In the 2022 post-COVID-19 study, the percentage of people between the ages of 50 and 59 more than doubled compared to the 2017 study (33.4% versus 15.8%). However, the percentage of people over 60 years old decreased by more than half, with 8.9% in 2022 compared to 18.9% in 2017. These figures indicate that older people were afraid to travel and venture outside their usual environment.

The motivation for travel remains primarily the same, that is, to experience the ancient olive trees and the oil production process, with 53.7% (in 2017) compared to 40.1% (in 2022), although in recent years there has been a growing interest in traveling to explore the local gastronomy [55].

Regarding the origin of tourists, nearly 60% are from the Andalucía region (in 2022) compared to 48.9% in 2019, although there is a slight increase in tourists from the European Union (6.8% in 2017, 10.3% in 2022).

In terms of visitor satisfaction, oleotourists are characterized as highly satisfied tourists, with a 95.4% satisfaction rate in 2022 compared to 83.2% in 2017, possibly due to the unique experience of visiting an olive grove, which is primarily found in Mediterranean Basin countries. Therefore, the current profile of the oleotourist is a male (57.2%), between 30 and 39 years old (26.8%), with secondary education (46.9%), originating from Andalucía (59.1%), with a monthly income level of EUR 1501–2000 (29.8%), spending less than 24 h in the area (56.5%), and spending between EUR 30 and 65 on the trip (20.9%), often accompanied by their partner (47.8%). They travel to explore the culinary traditions of the region, including the use of olive oil (49.5%), and they express high satisfaction with the trip and the experiences related to the world of olives.

**Table 4.** Comparison of the oleotourist profile in the years 2017, 2019, and 2022. Source: Own elaboration.

| Characteristics | | | Percentajes Study 2017 (564 Surveys) | Percentajes Study 2019 Pre-COVID-19 (630 Surveys) | Percentajes Study 2022 Post-COVID-19 (416 Surveys) |
|---|---|---|---|---|---|
| Personal data | Sex | Female | 43.3 | 42.3 | 42.8 |
| | | Male | 56.7 | 57.7 | 57.2 |
| | Age | Between 18 and 29 years | 17.4 | 14.4 | 16.1 |
| | | Between 30 and 39 years | 25.6 | 27.3 | 26.0 |
| | | Between 40 and 49 years | 22.3 | 19.7 | 15.6 |
| | | Between 50 and 59 years | 15.8 | 30.5 | 33.4 |
| | | More than 60 years | 18.9 | 8.1 | 8.9 |
| | Level of studies | No competed studies | 1.3 | 9.5 | 8.2 |
| | | Primary education | 25.2 | 20.5 | 18.3 |
| | | Secondary/VET studies | 59.3 | 42.0 | 46.9 |
| | | High level | 14.2 | 28.0 | 26.7 |
| | Marital status | Single | 27.1 | 27.3 | 27.6 |
| | | Married | 51.4 | 47.3 | 47.4 |
| | | Divorced/separated | 21.4 | 25.2 | 24.8 |
| | | Other | 0.1 | 0.2 | 0.2 |
| | Provenance | Andalucía | 48.9 | 59.4 | 59.1 |
| | | Rest of Spain | 42.3 | 29.8 | 29.6 |
| | | Rest of European Union | 6.8 | 10.2 | 10.3 |
| | | Rest of world | 2.0 | 0.5 | 0.7 |
| | Income | Less than EUR 1000 | 37.6 | 19.8 | 19.5 |
| | | EUR 1001–1500 | 12.6 | 20.0 | 19.7 |
| | | EUR 1501–2000 | 17.4 | 30.0 | 29.8 |
| | | EUR 2001–2500 | 14.2 | 20.0 | 20.7 |
| | | More than EUR 2500 | 18.2 | 10.2 | 10.3 |
| Itinerary | Number of days | Less than 24 h | 62.7 | 54.2 | 56.5 |
| | | Between 2 and 3 days | 29.1 | 33.3 | 36.1 |
| | | More than 3 days | 8.2 | 12.5 | 7.5 |
| | Average daily expenses | Less than EUR 30 | 27.2 | 11.1 | 13.9 |
| | | Between EUR 30 and 65 | 30.5 | 23.9 | 20.9 |
| | | Between EUR 66 and 100 | 27.6 | 43.9 | 44.0 |
| | | More than EUR 100 | 14.7 | 21.1 | 21.2 |
| | Individual accompanying | Alone | 17.9 | 2.0 | 2.0 |
| | | Accompanied my partner | 24.8 | 49.4 | 47.8 |
| | | Family | 15.3 | 38.4 | 10.1 |
| | | Friends | 42.0 | 10.2 | 35.8 |
| | Motivation | Oil mill visits | 53.7 | 50.2 | 40.1 |
| | | Culinary tradition region | 36.4 | 39.7 | 49.5 |
| | | Food festivals | 9.9 | 10.1 | 10.3 |
| | Satisfaction with the destination | Satisfied | 83.2 | 98.3 | 95.4 |
| | | Indifferent | 10.7 | 1.1 | 3.1 |
| | | Not satisfied | 6.1 | 0.6 | 1.4 |

## 5.2. Bivariate Analysis

Relationships have been established between various variables, such as gender, age, income level, and satisfaction level. The hypotheses are as follows:

**H0:** *The variables are independent.*

**H1:** *The variables are related.*

Table 5 shows the results of the bivariate analysis, where it can be observed that satisfaction with the trip to the world of olives is related to gender. Women are more satisfied than men when undertaking this type of tourism ($\chi^2$ = 17.896, Prob = 0.00013), similar to the findings of Mazon et al. [56] in their study of gastronomic tourism in the case of Benidorm. In that study, the satisfaction level among women was above 75% compared to 64% among men.

**Table 5.** Bivariate analysis.

| | $\chi^2$ | Degrees of Freedom | Prob | Accepted Hypothesis |
|---|---|---|---|---|
| Gender–satisfaction | 17.896 | 2 | 0.00013 * | $H_1$ |
| Age–satisfaction | 24.3521 | 8 | 0.002 * | $H_1$ |
| Income level–satisfaction | 11.4203 | 8 | 0.179 | $H_0$ |
| Motivation–satisfaction | 37.2498 | 4 | <0.00001 * | $H_1$ |
| Educational level–satisfaction | 7.42219 | 6 | 0.28356 | $H_0$ |
| Knowledge of olive oil–olive oil use | 24.7176 | 9 | 0.0033 | $H_1$ |

* Prob < $\alpha$ = 0.05.

The age variable is also related to satisfaction ($\chi^2$ = 24.3521, Prob = 0.002), as younger tourists are more satisfied than older ones. This differs from the results obtained in the studies by Pulido et al. [57], as living in cities, they are not familiar with many traditional methods of olive oil production and are more attracted to visiting old oil mills (Figure 8).

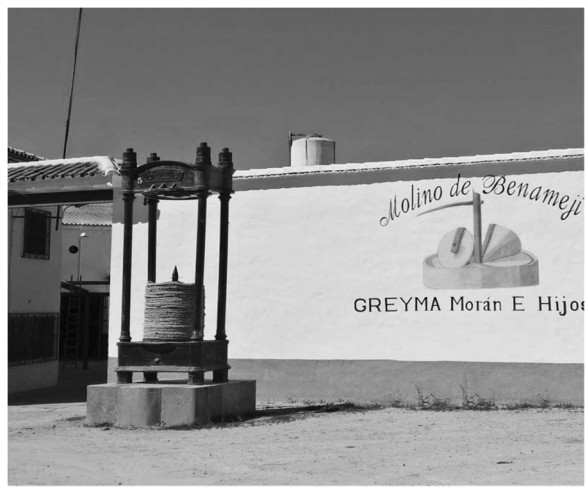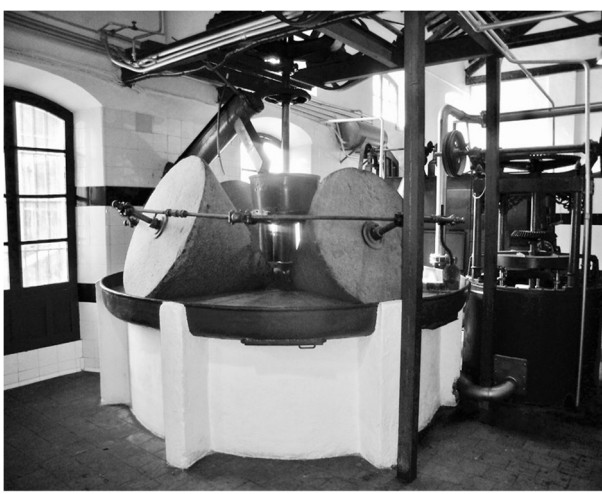

**Figure 8.** Benameji Olive Mill and olive press. Source: Authors.

There is no significant influence of income level on satisfaction ($\chi^2$ = 11.4203, Prob = 0.179), although there is a strong relationship between the knowledge of olive oil and its use (Figure 8). The type of oleotourist (oil novice, oil interested, oil lover, and oil connoisseur) is strongly associated with the type of oil used and the frequency of its use in cooking. Those interested in oil, who constitute the majority of tourists, use virgin olive oil several times a week. Connoisseurs (experts) use it daily and prefer higher-quality Extra Virgin Olive Oil (Figure 9).

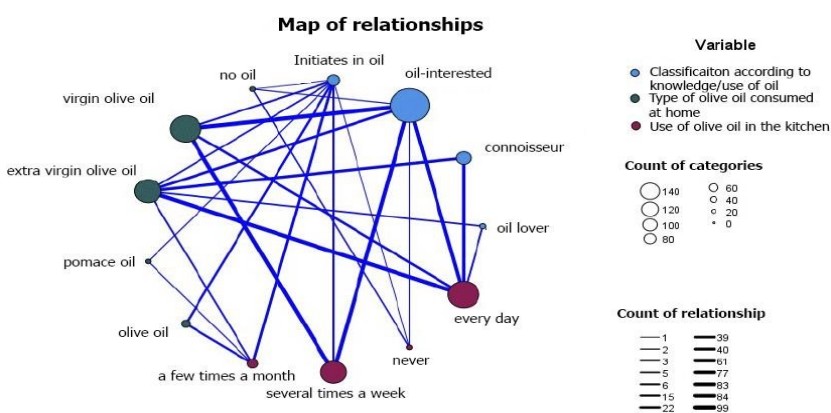

**Figure 9.** Map of relationships: Type of oleotourist—type oil used—how often the oil is used. Source: Authors.

The motivation for the trip is related to satisfaction, as individuals whose motivation is to learn about the olive culture are more satisfied than tourists whose main motivation is to explore the gastronomy. In general research on gastronomic tourism, which encompasses various products and not just oleotourism, it has been found that tourists who are more knowledgeable about the topic (those who chose gastronomy as the main motivation for their destination) have a higher level of satisfaction compared to tourists who selected gastronomy as a secondary motive for their trip. It is hypothesized that organized gastronomy tours function smoothly. Gastronomic tourists have specific needs, and tour organizers, based on their experience, know what tourists expect and require. Additionally,

gastronomic tourists rely on personal rather than commercial sources of information, which minimizes the gap between their expectations and experiences [58].

The level of education is not related to satisfaction ($\chi^2$ = 7.42219, Prob = 0.28356). These results differ from the research on gastronomic tourism by Dancausa and Millan [59], where the variables of education level and satisfaction were related. In their study, tourists with a higher level of education had slightly lower satisfaction, possibly because they believed that the explanations of the production process were not very precise or because they felt that more audiovisual resources were needed.

### 5.3. Comparison of Means Test

Various means comparison tests were conducted between numerical variables such as age, income level, average daily expenditure, and number of days spent in the area between the 2019 and 2022 studies to determine if the COVID-19 pandemic had affected the tourist profile (Table 6). However, no significant differences were found, as indicated by the probabilities of the Student's *t*-test being greater than the significance level of 0.05. This confirms the null hypothesis ($H_0$), indicating that the means before and after the pandemic are equal. In other words, the average daily expenditure and the number of days spent in the area by tourists remain unchanged before and after the COVID-19 pandemic, as well as their average income level and age.

**Table 6.** Results of mean comparison tests.

| Variable | Statistic | Prob | Accepted Hypothesis |
|----------|-----------|------|---------------------|
| Income | t = −0.2278 | 0.8198 | $H_0$: $\mu_1 = \mu_2$ |
| Age | t = −0.0157 | 0.8753 | $H_0$: $\mu_1 = \mu_2$ |
| Average Expense | t = 0.5360 | 0.5360 | $H_0$: $\mu_1 = \mu_2$ |
| Days | t = 1.2775 | 0.2017 | $H_0$: $\mu_1 = \mu_2$ |

Null Hypothesis $H_0$: $\mu_1 - \mu_2 = 0$, the means are equal before and after the COVID-19 pandemic. Alternative Hypothesis $H_1$: $\mu_1 - \mu_2 \neq 0$, the means are different before and after the COVID-19 pandemic.

Therefore, based on the results obtained, we can conclude that the COVID-19 pandemic has not changed the profile of oleotourism consumers. However, based on conversations with olive oil mill representatives, they have indicated that they have observed a change in the behavior of the tourists who visit them. Oleotourists are now more concerned about sustainability and more strongly adhere to safety and hygiene measures. Additionally, there has been an increase in the number of oleotourists after the pandemic (Figure 10).

### 5.4. SARIMA Demand Estimation Model

Studies on the evolution of inland tourism indicate that after the COVID-19 pandemic, tourists have sought contact with nature and have preferred to visit destinations near their place of residence. This implies that the world of olive tourism can be an attractive destination to be visited. However, a major challenge is that olive oil mills and olive landscapes, being located in rural areas, often lack the necessary infrastructure to meet the demands of olive tourists, such as hotels of a certain category, restaurants, and leisure activities related to the olive industry. To meet all these expectations, investment is needed in olive oil mills, creating olive museums, establishing trails to visit ancient olive trees, oil routes, etc. Therefore, it is necessary to understand the evolution of the demand for olive tourism in order to determine if investing in this tourism segment will be profitable or not.

There are few studies on the forecasting of the olive tourism demand, which highlights the importance of this study. To carry out this study, monthly information was collected from January 2009 to February 2023 (Figure 11) regarding the number of olive tourism visitors received by the olive oil mills open to the public in Andalucía. The variable oleotourists "number of olive tourists" was modeled using the Box–Jenkins methodology, specifically an ARIMA seasonal model, which analyzes a variable based on its past values.

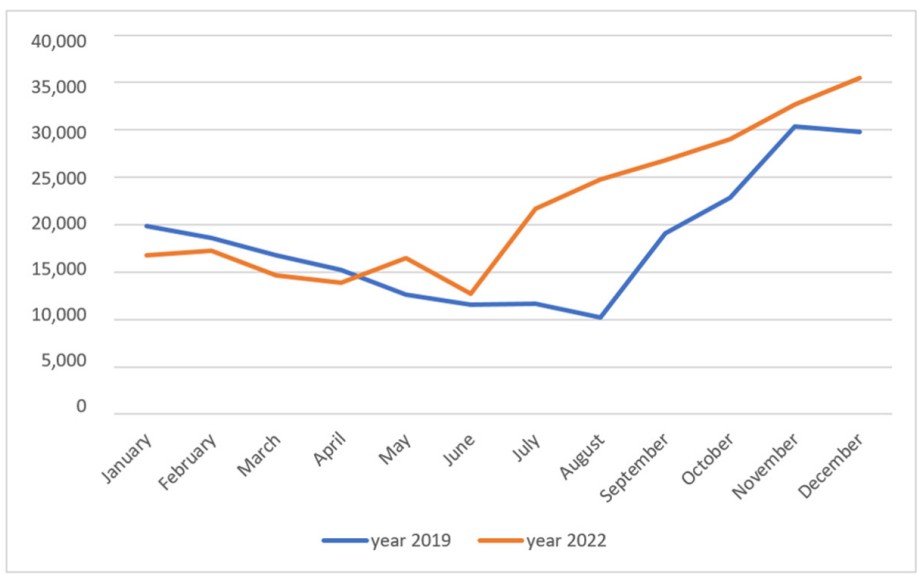

**Figure 10.** Evolution of the number of oleotourists in Andalucía before the COVID-19 pandemic in 2019 and post-COVID-19 in 2022.

$$\Phi(B)\,\phi(B)\,(1-B)d\,(1-Bs)D\,Y(\lambda t = \theta(B)\,\theta(B)\,at$$

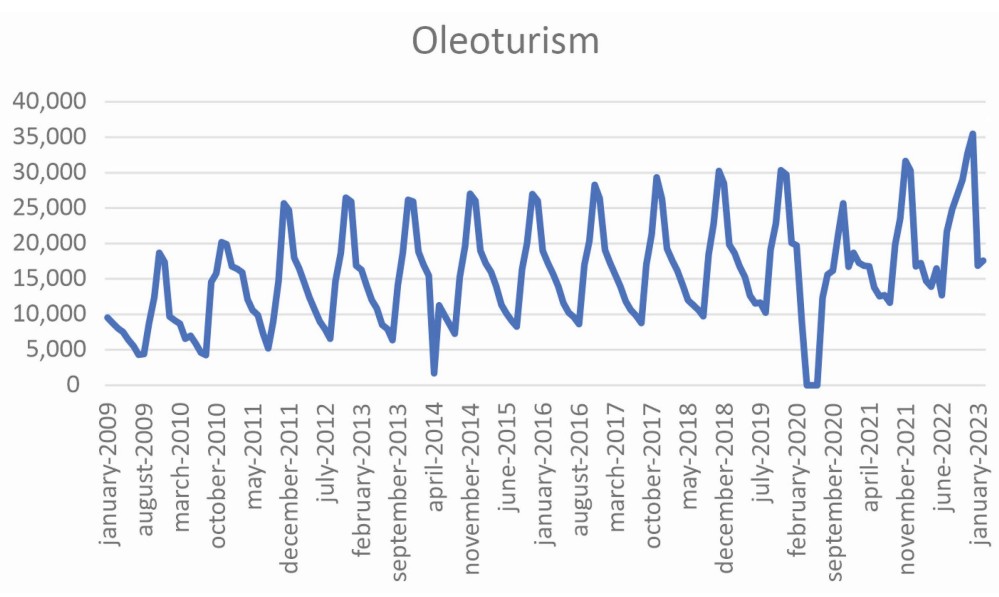

**Figure 11.** Monthly evolution of the number of olive tourism visitors in Andalucía (January 2009 to February 2023). Source: Own elaboration. Vertical axis in thousands of oleotourists.

The demand for olive tourism in Andalucía, referred to as "oleotourism" in the model, is a variable with a trend in the variance, which has been corrected by applying the Box–Cox transformation with $\lambda = 0.2$ (Oleoturism1 = oleoturism^0.2). It also exhibits a trend in the mean and cycle, which have been addressed through differencing in the mean and cycle.

The estimated model for the monthly forecast of the olive tourism demand in Andalucía is presented in Table 7, and it is a SARIMA $(1, 1, 0)\,(0, 1, 1)_{12}$ model.

$$(1 + 0.532820\,B)\,(1 - B^1)\,(1 - B^{12})^1\,Oleoturism^{0.2} = at$$

$t_{\phi 1} = -7.542716$ * Significant coefficient.

**Table 7.** Estimation of the demand for oleotourism in Andalucía.

| Dependent Variable: D(OLEOTURISM^0.2,1,12) Method: Least Squares | | | | |
|---|---|---|---|---|
| Variable | Coefficient | Std. Error | t-Statistic | Prob. |
| AR(12) | −0.532820 | 0.070640 | −7.542716 | 0.0000 |

Table 8 shows the results of the Dickey–Fuller test. Since the probability (0.4805) is greater than $\alpha = 0.05$, the null hypothesis $H_0$ that the variable "oleotourism" has a unit root is satisfied.

**Table 8.** Unit root test.

| Null Hypothesis: OLEOTURISM Has a Unit Root Exogenous: Constant | | | |
|---|---|---|---|
| Lag Length: 12 (automatic—based on SIC, maxlag = 13) | | t-Statistic | Prob. * |
| Augmented Dickey–Fuller test statistic | | −1.599346 | 0.4805 |
| Test critical values: | 1% level | −3.472259 | |
| | 5% level | −2.879846 | |
| | 10% level | −2.576610 | |

* the variable "oleotourism" has a unit root.

Table 9 reflects the results of the ARCH test. Since the probability (0.8559) is greater than $\alpha = 0.05$, the null hypothesis $H_0$ of the absence of autoregressive conditional heteroskedasticity is satisfied.

**Table 9.** Heteroskedasticity test: ARCH.

| Heteroskedasticity Test: ARCH | | | |
|---|---|---|---|
| F-statistic | 0.032529 | Prob. F(1,142) | 0.8571 |
| Obs * R-squared | 0.032980 | Prob. Chi-Square(1) | 0.8559 |

* Absence of autoregressive conditional heteroskedasticity.

With this estimated SARIMA model, monthly predictions have been made for the year 2024, and the results are presented in Table 10. It can be deduced that the evolution of oleotourism shows a growing trend (22.57%) over these 7 years (2017–2024), with an increase of 45,316 people compared to the year 2017. This indicates that this form of tourism is attractive to potential tourists. However, there is a pronounced seasonality in the months of October, November, and December, which comprise the olive harvesting and processing season. During this time, oleotourism becomes more appealing as visitors can witness the full olive oil production process in action.

**Table 10.** Forecast for rural tourism demand in Andalucía for 2024 (persons).

| | Year 2017 | Year 2024 |
|---|---|---|
| February | 17,245 | 17,489 |
| March | 15,538 | 16,725 |
| April | 13,956 | 14,983 |
| May | 11,864 | 12,762 |
| June | 10,635 | 16,475 |
| July | 9846 | 13,984 |
| August | 8764 | 25,435 |
| September | 17,023 | 26,978 |
| October | 21,269 | 29,102 |
| November | 29,325 | 33,675 |
| December | 26,245 | 38,469 |
| Total | 200,761 | 246,077 |

If the olive landscape is declared a World Heritage Site, the demand for tourists is likely to increase even further, as the olive trees can be visited throughout the year and not just during the harvest season. This would help to diversify the demand and create wealth for Andalucía.

## 6. Discussion

The landscape of the olive groves has the potential to become a tourist attraction for rural areas in Andalucía, attracting numerous tourists if it is declared a World Heritage Site, similar to other landscapes like the agave landscape in Mexico. According to Danzan and Gonzalez [60], the declaration of a World Heritage Site in Mexico led to an increase in the number of tourists, generating benefits such as employment diversification, local cultural recovery, and the emergence of new businesses. However, it is important to address the negative effects, including increased crime rates, pollution, and limited distribution of tourism benefits to local communities, with most benefits going to foreign tour operators.

If the olive grove landscape is declared a World Heritage Site, it can generate additional income for farmers [61]. However, it is crucial to ensure that tourism does not negatively impact the environment by exceeding the carrying capacity of rural areas, where the negative effects outweigh the benefits.

To determine whether the olive grove landscape meets the requirements for World Heritage Site designation, it is essential to consider the parameters that UNESCO assigns high value to, such as historical characteristics, traditional crops and local products, land use continuity, agricultural practices, and the presence of architecture related to agricultural activities [62]. Fortunately, these parameters align with the olive grove landscape.

If Andalucía possesses a distinctive landscape characterized by a "sea of olive trees", it should not only benefit the agricultural sector but also the tertiary service sector, such as tourism. Landowners are well aware of the contribution of nature to their livelihoods and lifestyles. They are concerned about rural exodus and the encroachment of shrublands, which could negatively impact the socio-ecological context they value and depend on. The main needs of landowners to support biodiversity conservation are not primarily driven by economic interests but are more related to the necessity for support that can enhance land management and social cohesion in Uruguay [63].

Oleotourism, a tourism activity related to the world of olive oil (including landscapes, olive oil production processes, and the "green gold" extracted from olives), has gained momentum after the COVID-19 pandemic. Spanish tourists have shifted their vacation habits from crowded areas to rural destinations, benefiting the rural areas of Andalucía. Although the profile of oleotourists has not changed significantly before and after the pandemic, it has been observed that the motivation for an authentic territorial experience and learning about local customs and sustainable development practices has become increasingly attractive to tourists. This has led to an increase in the number of tourists. The integration of olive mills and local businesses into oleotourism routes (including restaurants, shops, and rural accommodations) promotes and preserves the natural environment in a respectful manner, as observed in the studies by Dàuria et al. in Croatia, Italy, and Spain [64].

Analyzing tourist behavior, it is evident that Andalucía and its olive groves provide an ideal context for the promotion of initiatives related to oleotourism, as shown in the studies by Campon et al. [65]. These authors collected primary data from a sample of British tourists, and their analysis revealed an interest in Spain, olive oil, and engaging in activities related to oleotourism in the region of Extremadura, which can be extrapolated to Andalucía.

## 7. Conclusions

The landscapes of Andalucía, including the dehesa with its dominant holm oaks, whose acorns are the main food source for Iberian pigs, and the olive groves, from which olive oil is obtained, constitute a cultural resource consisting of tangible and intangible

heritage that can be potentially exploited from a tourism perspective. In this study, we have analyzed the culture of the olive groves by investigating the evolution of olive oil tourism, particularly the profile of oleotourists before the COVID-19 pandemic in 2019 and after the pandemic in 2023. It is observed that the personal characteristics of oleotourists remain similar, predominantly consisting of local tourists, males aged between 40 and 50 years, with a medium income level of EUR 1500–2000, and who do not usually stay overnight in the area. Their average expenditure ranges between EUR 60 and 100, primarily invested in purchasing oil from the visited olive mills.

It has been observed that this type of tourism has increased in number since the COVID-19 pandemic, along with the motivations of tourists, as they seek experiential and sustainable tourism experiences. This research advocates for considering the declaration of the olive grove landscape as a World Heritage Site, as it would benefit the rural areas where it is located by increasing agricultural income through olive tourism, especially in years with poor olive harvests. However, there is still a long way to go, as creating a quality tourism product goes beyond having high-quality raw materials such as oil, olive fields, ancient olive trees, olive mills, gastronomy, etc. It requires the creation of tourism products based on the profile and segmentation of oleotourists, as pointed out by Milla et al. [35].

This study has practical and managerial implications. The identified segments allow for a better understanding of the profile of tourists visiting the Andalucía region based on their level of involvement. An important practical contribution of this research is that not all tourists have the same degree of involvement, which is dependent on their knowledge of oil and its daily use. This finding has significant implications for heritage destinations, as both less-involved and highly involved tourists share a similar interest in the physical environment, particularly the traditional oil landscape.

Our findings can help local governments and businesses associated with the oil industry adapt their marketing strategies for the destination. Differentiating between highly involved and less-involved tourist segments also contributes to identifying the most relevant factors that could guide public–private partnerships to protect and promote the World Heritage destination.

While different organizations and institutions in the olive industry are promoting their own products, it is recommended that the industry considers working cooperatively to communicate with the general public about the benefits of visiting the olive grove region. This includes creating a common brand and implementing various marketing campaigns based on target markets. The collaboration of all stakeholders, including olive producers, tourism sector entrepreneurs, public organizations, etc., can enhance the value of the olive grove landscape in national and international markets, saving costs in promoting Andalucía as a cultural, gastronomic, rural, and sustainable tourist destination. The multiculturalism in the region has influenced the evolution of the olive grove world, as traditions from different peoples who inhabited the region throughout the centuries have blended together, creating a forest of olive trees and an Extra Virgin Olive Oil appreciated by all palates that taste it.

In conclusion, designating the olive grove landscape a World Heritage Site in Andalucía would benefit the agricultural community by preserving and promoting this cultural practice and ensuring its long-term sustainability. It would also promote sustainable tourism in Andalucía, allowing visitors to learn about the history and production of olive oil while protecting the natural environment and supporting local communities.

**Author Contributions:** Conceptualization, M.G.D.M., J.S.-R.G. and M.G.M.V.d.l.T.; methodology, M.G.D.M.; software, J.S.-R.G. and M.G.D.M.; validation, M.G.D.M. and M.G.M.V.d.l.T.; formal analysis, M.G.D.M. and M.G.M.V.d.l.T.; investigation, M.G.D.M. and M.G.M.V.d.l.T.; resources, M.G.M.V.d.l.T. and J.S.-R.G.; data curation, M.G.D.M.; writing—original draft preparation, M.G.D.M. and M.G.M.V.d.l.T.; writing—review and editing, M.G.D.M. and J.S.-R.G.; visualization, M.G.D.M.; supervision, M.G.M.V.d.l.T. and J.S.-R.G.; project administration, M.G.D.M. and J.S.-R.G.; funding acquisition, M.G.M.V.d.l.T. All authors have read and agreed to the published version of the manuscript.

**Funding:** This research received no external funding.

**Informed Consent Statement:** Informed consent was obtained from all subjects involved in the study.

**Data Availability Statement:** The data presented in this study are available on request from the corresponding author.

**Conflicts of Interest:** The authors declare no conflict of interest.

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
