# Peer review of "The Olive Grove Landscape as a Tourist Resource in Andalucía: Oleotourism"

_land, doi:10.3390/land12081507_

Round 1

Reviewer 1 Report

land-2520570-peer-review-v1.

Review of “The olive grove landscape as a tourist resource in Andalucía:  Oleotourism”

This is an interesting paper that is worth publishing, but that needs a revision taking into account the comments made below.

The paper also has numerous typographic mistakes, left-over elements of Spanish (in figures and tables) and issues  in formatting which suggest careless or hasty manuscript preparation. One wonders, then, how careless or hasty the analysis of the data may have been as well..

First section. From a heritage perspective, cultural landscapes need not be agroecosystems. There are large landscapes that have been created by mining activity, which resulted in denuded hill sides (wood for fuel and stoping on mine tunnels). In Australia, for ex. Indigenous Australian peoples engaged in patchwork burning, that shaped the land scape, but that cannot be reads as an  Agroecosystem. Add 1-2 references o cultural landscapes here.

Line 71 ff         Cultural landscapes MAY reflect sustainable landuse, but there are many exploitative unsustainable cultural land scapes. Say “Some cultural landscapes…

Line 104ff        Make some commentary on the antiquity and longevity of these landscapes , reaching back several hundred years, with some olive trees older than 1000. This framing adds significance

Add a comment that the cultural landscape not only encompasses the olive groves, but also the rural communities and specific structures and equipment, and that the intangible heritage here relates to production techniques. From a cultural tourism perspective, more of this context is necessary.

I would consider splitting section 2 into the heritage context and the tourism aspects (two subheadings)

Line 283 ff       “wild groves” … the trees are clearly planted in rows, so these are not ‘wild’ (ie self seeded), these are ABANDONED groves … 

Line 461          give reference to changes consumer behaviour

Table 3                       Tourists of both sexes over 18 …what about gender diverse people?

Line 782          “Institutional Review Board Statement: Not applicable”  The authors need to provide a formal determination by the Ethics Review Board of their university that a formal Ethics approval was not required. In this day and age of academic ethical scholarhip this will be required prior to publication.

MINOR ISSUES

Line 52            UNESCO World Heritage Convention, 1972 not 1992

Figure 1            The text in the bars needs to be in English

Table 1 In English, the decimal signifier (for the percentages) is a full stop , not a comma so: 60.38  not 60,38 …

Line 486 Date of fieldwork         Dates need to be in English

Figure 5 has no frame, figure 6 has a frame. Be consistent

Table 4  The layout is really messy, needs to be fixed

Line 627            Spelling mistake ‘onthly’

Figure 10          The bottom axis needs to be converted into month-year to be readable

Line 675            missing end bracket

LANGUAGE

Some minor infelicities in expression.

Example: “Own elaboration based on data from the”

Line 356          avoid colloquial English such as “We are talking about”

Line 612;         “Comparison of the evolution of”  “evolution” ?

LANGUAGE

Some minor infelicities in expression.

Example: “Own elaboration based on data from the”

Line 356          avoid colloquial English such as “We are talking about”

Line 612;         “Comparison of the evolution of”  “evolution” ?

Author Response

Dear Reviewer:

We appreciate your feedback on the article.

We have tried to address all your comments and suggestions.

First section. From a heritage perspective, cultural landscapes need not be agroecosystems. There are large landscapes that have been created by mining activity, which resulted in denuded hill sides (wood for fuel and stoping on mine tunnels). In Australia, for ex. Indigenous Australian peoples engaged in patchwork burning, that shaped the land scape, but that cannot be reads as an Agroecosystem. Add 1-2 references o cultural landscapes here.

We have added references about the mining landscape and the Amazonia landscape not related to agricultural activity."

Line 71 ff Cultural landscapes MAY reflect sustainable landuse, but there are many

The word 'some' has been added.

Line 104ff Make some commentary on the antiquity and longevity of these landscapes , reaching back several hundred years, with some olive trees older than 1000. This framing adds significance Add a comment that the cultural landscape not only encompasses the olive groves, but also the rural communities and specific structures and equipment, and that the intangible heritage here relates to production techniques. From a cultural tourism perspective, more of this context is necessary.

The paragraph about the antiquity of the olive grove and ancient olive trees has been added

I would consider splitting section 2 into the heritage context and the tourism aspects (two subheadings)

It has been preferred to keep it in the same section to link the olive grove with tourist activity."

Line 283 ff “wild groves” … the trees are clearly planted in rows, so these are not ‘wild’ (ie self seeded), these are ABANDONED groves …

 The term 'wild olive groves' has been changed to 'abandoned olive groves'

… Line 461 give reference to changes consumer behaviour Table 3 Tourists of both sexes over 18 …what about gender diverse people? Table 3 .  Tourists of both sexes over 18 …what about gender diverse people?

The phrase has been modified, indicating only tourists.

Line 782          “Institutional Review Board Statement: Not applicable”  The authors need to provide a formal determination by the Ethics Review Board of their university that a formal Ethics approval was not required. In this day and age of academic ethical scholarhip this will be required prior to publication.

It is not necessary, as the participants were informed about the purpose of the study, and they provided their verbal consent. This has been stated in section 4 of materials and methods.

 “The access by the surveyors to the olive oil route/PDO/PGI and the conduct of interviews with tourists were authorized by the managing body and owner of the DOP's/PGI's. Prior to completing the questionnaire, tourists were informed of the academic purposes and the anonymity of their answers. Verbal consent was obtained to participate in the question-naire. The anonymity of the visitors to the olive oil route/PDO/PGI was guaranteed at all times. “

Line 52            UNESCO World Heritage Convention, 1972 not 1992

The date to which we want to refer is 1992.

"Cultural landscapes are briefly noted in a broad historical and intellectual context. They are examined inthe context of the World Heritage Convention (1972) and its application. The specific focus is between December, 1992, when the World Heritage Committee recognised ‘cultural landscapes’ as a category ofsite within the Convention's Operational Guidelines, and 30 June, 2002, at which point 30 World Heritage cultural landscapes had been officially recognised.

Figure 1            The text in the bars needs to be in English

The text in the bars has been changed to English

Table 1 In English, the decimal signifier (for the percentages) is a full stop , not a comma so: 60.38  not 60,38 …

Se ha cambiado la coma por el  punto en la tabla 1

The comma has been changed in Table 1

Line 486 Date of fieldwork         Dates need to be in English

The dates have been translated into English

Figure 5 has no frame, figure 6 has a frame. Be consistent

A frame has been added to Figure 6.

Table 4  The layout is really messy, needs to be fixed

It has been corrected

Line 627            Spelling mistake ‘onthly’

It has been corrected from 'onthly' to 'monthly

Figure 10          The bottom axis needs to be converted into month-year to be readable

The figure has been improved for better comprehension.

Line 675            missing end bracket

A bracket has been added.

Example: “Own elaboration based on data from the”

It has been corrected to 'own elaboration'

Line 356          avoid colloquial English such as “We are talking about a forest”

The expression has been corrected to 'It is a forest'

Line 612;         “Comparison of the evolution of”  “evolution” ?

The expression 'comparison of evolution' has been changed to 'evolution

Reviewer 2 Report

Dear Authors, thank you for the interesting manuscript!

The text needs some revisions. Please see more detailed comments in the text below:

¾    Lines 165-227 - "This plan is based on 11 objectives: ....." Is this long citation is a necessary for the manuscript? I would like to suggest rewriting these 11 items into summed up sentences.

¾    The map of Spain with Andalucía could increase the visibility of the investigated region.

¾    Could you give some quantitive and qualitative information about current tourist opportunities of the Andalucía region (accommodations, restaurants, other resources related to tourism). If you could present the data by provinces it would be increase the significance of your research.

¾    Lines 723-724 - Is this detailed information (whose acorns are the main food source for Iberian pigs) an important for the research?

¾    Please, check out the content of the Table 2

¾    Lines 755-766 - Please, add the detailing word "olive" into the sentences

¾    Please, check the text for translations some Spanish words, for instance, Fig. 1., Table 4., dehesa.

¾    Please, check the text for misspelled words, extra letters and punctuations (lines 493, 502, 520, hours in Table 4, Monthly in Fig. 10, 635).

 This is an interesting research, which brings in new empirical findings. It thus has a clear merit.

Author Response

Dear Reviewer:

We appreciate your feedback on the article.

We have tried to address all your comments and suggestions.

the text needs some revisions. Please see more detailed comments in the text below:

  • Lines 165-227 - "This plan is based on 11 objectives: ....." Is this long citation is a necessary for the manuscript? I would like to suggest rewriting these 11 items into summed up sentences.

Some of the 11 objectives have been simplified.    

  • The map of Spain with Andalucía could increase the visibility of the investigated region.

A map of Spain has been included, showing Andalusia.

  • Could you give some quantitive and qualitative information about current tourist opportunities of the Andalucía region (accommodations, restaurants, other resources related to tourism). If you could present the data by provinces it would be increase the significance of your research.

A paragraph with quantitative information about the tourism sector in Andalusia has been added

  • Lines 723-724 - Is this detailed information (whose acorns are the main food source for Iberian pigs) an important for the research?

It has been indicated that the "dehesa" is a type of cultural landscape and a source of food for the Iberian pig, from which the Iberian ham is obtained, which is another gastronomic product that is beginning to be exploited as a tourist product.

  • Please, check out the content of the Table 2

It has been corrected the text of table 2.

  • Lines 755-766 - Please, add the detailing word "olive" into the sentences

It has been added the word "olive" in the sentences indicated by the reviewer.

  • Please, check the text for translations some Spanish words, for instance, Fig. 1., Table 4., dehesa.

The words that appeared in Spanish in Figure 1 and Table 4 have been translated, and the word "dehesa" remains the same in English.

  • Please, check the text for misspelled words, extra letters and punctuations (lines 493, 502, 520, hours in Table 4, Monthly in Fig. 10, 635).

The errors indicated by the reviewer have been corrected

Round 2

Reviewer 1 Report

The authors have done well the revise their submission. Some  minor issues remain:

Line 52            UNESCO World Heritage Convention, 1972 not 1992

The date to which we want to refer is 1992.

"Cultural landscapes are briefly noted in a broad historical and intellectual context. They are examined inthe context of the World Heritage Convention (1972) and its application. The specific focus is between December, 1992, when the World Heritage Committee recognised ‘cultural landscapes’ as a category ofsite within the Convention's Operational Guidelines, and 30 June, 2002, at which point 30 World Heritage cultural landscapes had been officially recognised.

I understand what you are saying, but as it stands its wrong. I suggest this:

essential element of world culture (UNESCO World Heritage Convention, 1972 (as amended 1992]; Mediterranean Landscape Charter, 1992;

====================

Figure 10          The bottom axis needs to be converted into month-year to be readable

The figure has been improved for better comprehension.

This has not been fixed changes the labels on the bottom axis from 09 10 11  to 2009, 2010 etc (the years can be aligned sideways)

Author Response

Reviewer 1

Dear Reviewer:

We appreciate your feedback on the article.

We have tried to address all your comments and suggestions.

I understand what you are saying, but as it stands its wrong. I suggest this:

essential element of world culture (UNESCO World Heritage Convention, 1972 (as amended 1992]; Mediterranean Landscape Charter, 1992;

The change suggested by the reviewer has been made

Figure 10          The bottom axis needs to be converted into month-year to be readable

This has not been fixed changes the labels on the bottom axis from 09 10 11  to 2009, 2010 etc (the years can be aligned sideways)

The change suggested by the reviewer has been made
